# Selection of Ground Motion Intensity Measures and Evaluation of the Ground Motion-Related Uncertainties in the Probabilistic Seismic Demand Analysis of Highway Bridges

**Huihui Li** [1,2] **, Guojie Zhou** [3] **and Jun Wang** [1,2,*]

1 College of Civil and Transportation Engineering, Shenzhen University, Shenzhen 518060, China
2 Guangdong Provincial Key Laboratory of Durability for Marine Civil Engineering, Shenzhen University, Shenzhen 518060, China
3 State Key Laboratory of Coastal and Offshore Engineering, Department of Civil Engineering, Dalian University of Technology, Dalian 116023, China
* Correspondence: wangjun123@szu.edu.cn

**Abstract:** Probabilistic seismic demand analysis (PSDA) is known as one of the critical intermediate steps in the performance-based earthquake engineering (PBEE) design framework. Accuracy of the PSDA is influenced by various sources of uncertainties and mostly by that stemming from the ground motion-related variabilities. By taking a representative reinforced concrete (RC) continuous girder bridge as the case study, twenty-eight commonly used seismic intensity measures (IMs) were investigated in terms of the effectiveness, efficiency, practicality, proficiency, and sufficiency assessments. Probabilistic seismic demand models (PSDMs) of several critical bridge engineering demand parameters (EDPs) were developed under both the near-field and far-field ground motions through the nonlinear time history analyses (NTHAs). In addition, effects of ground motion-related uncertainties, such as the bin-to-bin (BTB) and record-to-record (RTR) variabilities, on the PSDA of highway bridges were also investigated. It is concluded that (1) IM efficiency contributes significantly to reflecting the RTR variability of ground motions and an efficient IM may reduce the influence of RTR variability in the estimation of structural demands; (2) IM sufficiency reflects the statistical independence of IM and ground motion parameters, and a sufficient IM is helpful in rendering the prediction of structural demands; and (3) uncertainties stemming from both the BTB and RTR variabilities of the seismic records have significant influences on the PSDA and the developed PSDMs of highway bridges.

**Keywords:** intensity measure (IM); PSDA; PSDMs; IM selection; ground motion-related uncertainties

## 1. Introduction

In the current PBEE design framework, there are mainly the following four crucial parts: (i) seismic hazard analysis, (ii) seismic response analysis, (iii) seismic damage analysis, and (iv) seismic loss estimation [1,2]. In this design framework, the structural seismic responses and demands can usually be predicted by using the PSDA through the NTHAs [2–7]. Based on the PSDA, the structural demands are often characterized by the developed PSDMs, which illustrate the predicted seismic demands with respect to the given ground motion IMs [2–7]. The PSDM anticipates the structural demands and provides the conditional probability that a structural EDP meets or exceeds a certain value ($D$), which can be represented as $P[EDP \geq D \mid IM]$ [2].

Accuracy of the PSDA significantly relies on the uncertainty level involved in the developed PSDMs, which, in turn, rely on the selection of ground motion IMs. Thus, the appropriate selection of IMs will reduce the dispersion of the developed PSDMs, and then lead to more reliable structural response and demand predictions [2]. In this regard, many previous studies [2,8–12] have contributed to the investigations on the evaluation of the

IM selections based on several critical metrics for bridge structures and buildings. These critical evaluation metrics of the IM selection include effectiveness, efficiency, practicality, proficiency, sufficiency, and hazard computability [2,12]. For instance, based on the above-mentioned metrics, Padgett et al. [10] performed the IM selection of ten seismic IMs for some bridges in the United States, and they suggested that the peak ground acceleration (*PGA*) was the most suitable IM for the PSDA of highway bridges. Bradey and Cubrinovski [13] conducted the selection of IMs for pile foundations with both liquefiable and non-liquefiable soils, and they suggested that the velocity spectrum intensity (*VSI*) was suitable to predict the pile's response. Similarly, by taking an extended pile-shaft-supported bridge as the case study, Wang et al. [14] compared the optimal selection of twenty-six commonly employed seismic IMs, and they found that the velocity-related IMs may contribute to more reliable PSDMs for the studied extended pile-shaft-supported bridges compared to the time-related, displacement-related, and acceleration-related IMs. Likewise, to extend the applications of the current metrics for the IM evaluation, Khosravikia and Clayton [2] proposed several alternative solutions to study the evaluation metrics of the IM efficiency, practicality, and proficiency to reduce the influence of uncertainty levels on the structural demand parameters. They also found that the velocity-related seismic IMs (e.g., *PGV*) were suitable for steel girder bridges in the United States.

Furthermore, in the PBEE design framework, it is vital to develop some possible techniques that can take into consideration the uncertainties involved in the structural seismic response, seismic demand, and seismic fragility assessments [12]. For example, the structural seismic demands should be accurately evaluated to obtain the reliable seismic vulnerability, seismic damage, and loss estimation. However, uncertainties in the seismic responses and demands derived from variabilities in the input parameters that are related to the structural modeling and/or ground motions may decrease this accuracy [12]. There are a number of sources of uncertainties, i.e., the structure-to-structure (STS) (i.e., material, geometric information), the bin-to-bin (BTB) (i.e., uncertainties involved in different selected ground motion databases, such as the far-field ground motion database via the near-field ground motion database in the present study), and record-to-record (RTR) variabilities of earthquake records, due to the related uncertain parameters in material, geometric, and structural properties, modeling assumptions, static or dynamic loadings, and selection of the input seismic records [3–7,15–17]. Moreover, according to Kiureghian and Ditlevsen [18], uncertainties that are involved in earthquake engineering may be categorized into two different aspects: (i) the aleatory and (ii) the epistemic uncertainties. The former mainly comes from the STS, RTR, and BTB variabilities, whereas the latter mainly comes from the lack of statistical data and human knowledge [3–7]. In this regard, we may either ignore the contribution of crucial uncertain parameters to the predicted PSDMs and seismic risk and vulnerability analyses of bridge structures; or, conversely, we may put much unnecessarily effort into complicated simulations that are less helpful in predicting the seismic responses and the PSDA of structures [3–7,15,16]. Thus, it is crucial and fundamental to study the effects of the STS, BTB, and RTR variabilities on the seismic responses of structures. Therefore, the readers may refer to the authors' previous studies [5,6] regarding the investigations on the influences of uncertainties derived from STS variability on the seismic responses and vulnerability assessments of highway bridges, while the present study is trying to investigate the effects of uncertainties coming from the BTB and RTR variabilities on the PSDA of highway bridges.

On the one hand, many previous studies mainly focused on the optimal selection of ground motion IMs when the considered structures were under far-field ground motions, whereas those under near-fault ground motions were relatively limited. On the other hand, most of the previous studies contributed to exploring the effects of uncertainties derived from STS variabilities, such as material-related, numerical modeling-related, and boundary condition-related uncertainties, on the seismic responses and vulnerability assessments of highway bridges, while very few studies have been dedicated to investigating the influences of ground motion-related uncertainties, such as the BTB and RTR variabilities,

on the seismic responses and seismic fragility estimates of highway bridges. To this end, the main objectives of the present study are to (i) investigate the proper evaluation of twenty-eight commonly utilized ground motion IMs in terms of their effectiveness, efficiency, practicality, proficiency, and sufficiency assessments for both the near-fault and far-field ground motions; and (ii) investigate the influences of the ground motion-related uncertainties, such as the BTB and RTR variabilities of both the near-fault and far-field ground motions, on the PSDA and the developed PSDMs for several critical bridge EDPs. Thus, the present study first mainly involved the procedure to develop the PSDM for a given EDP, and introduced the generally used evaluation criteria for the selection of the seismic IMs. Then, this paper presents the brief introductions of the numerical modeling of the case study bridge and several considered critical bridge EDPs. Subsequently, introductions of the twenty-eight seismic IMs and the fundamental information of the input far-field and near-fault ground motions are given. Detailed results and discussions about the ground motion IM selection for the considered IMs with respect to their evaluation metrics are presented. Moreover, this study involved a detailed investigation on the influences of ground motion-related uncertainties, such as the BTB and RTR variabilities, on the PSDA and the developed PSDMs of highway bridges.

## 2. Probabilistic Seismic Demand Model

PSDA can be applied to estimate the mean annual frequency ($v$) of a structure under a specific hazard ($IM > x$), exceeding a given structural EDP ($EDP > y$), which can be written as [12,19]

$$v_{EDP}(y) = \int_x G_{EDP|IM}(y|IM = x)|d\lambda_{IM}(x), \tag{1}$$

where $G_{EDP|IM}(y|IM = x)$ is the function model in predicting the conditional probability of a structural EDP for a given IM and $\lambda_{IM}(x)$ is the seismic hazard model in predicting the annual probability. Based on the studies by Khosravikia and Clayton [2] and Cornell et al. [20], a conditional PSDM generally follows a lognormal distribution, which can be expressed as

$$P(EDP \geq D|IM) = 1 - \Phi\left(\frac{\ln(D) - \ln(S_D)}{\beta_{D|IM}}\right), \tag{2}$$

where $\Phi$ is the standard normal cumulative distribution function; $S_D$ is the median structural seismic demand, and $\beta_{D|IM}$ is the logarithmic dispersion of the seismic demand conditioned on the seismic IM. Based on some previous studies [4,5,21–23], the structural demand and structural capacity of a specific bridge component generally follow the lognormal distributions [3–7,21,22], and the PSDM can be represented by

$$S_D = a \cdot IM^b \quad \text{or} \quad \ln(S_D) = \ln(a) + b \cdot \ln(IM). \tag{3}$$

Thus, as seen from the representative illustration of a PSDM shown in Figure 1, coefficients $a$ and $b$ can be obtained through the linear regression analysis. Furthermore, by assuming $S_D$ follows a lognormal distribution, the dispersion $\beta_{D|IM}$ of the developed PSDM can be calculated by [4,5,21–23]

$$\beta_{D|IM} = \sqrt{\frac{\sum_i^n \ln(D_i) - \ln(S_D)^2}{n-2}} = \sqrt{\frac{\sum_i^n \ln(D_i) - \ln(a \cdot IM^b)^2}{n-2}}, \tag{4}$$

where $n$ is the number of simulations, and $D_i$ represents the $i$th realization of the structural demand from the NTHAs. Hence, it is evident that the reasonable selection of seismic IMs is crucial to improve the capability of PSDMs to capture the structural seismic responses.

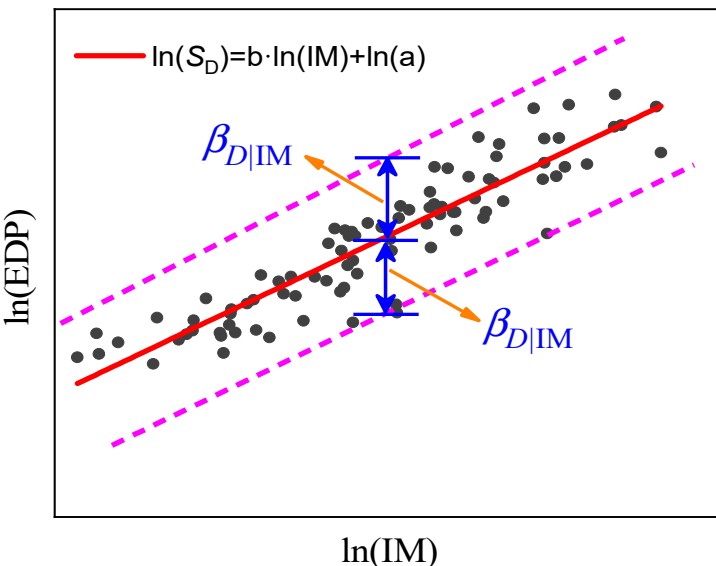

**Figure 1.** Illustration of a representative PSDM in natural log space.

## 3. Evaluation Criteria for the Optimal Ground Motion IMs

According to some previous studies [2,8–10,12,24], the following evaluation metrics are generally employed to select the seismic IMs, including (i) efficiency, (ii) practicality, (iii) proficiency, (iv) sufficiency, and (v) hazard computability. Thus, introductions of these criteria are briefly given in the following subsections.

### 3.1. Efficiency

Efficiency usually reflects the variation in the generated structural demand and it can be quantified by using $\beta_{D|IM}$ calculated in Equation (4). Generally, a more efficient IM tends to generate a lower value of $\beta_{D|IM}$, suggesting relatively less variation in the predicted structural demand from Equation (3). According to several previous studies [12,19,25], if $\beta_{D|IM}$ is in the range of 0.2~0.3, the selected IM can be considered as efficient; however, a range of 0.3~0.4 can be still considered as satisfactory. Moreover, to further ensure the rationality of the developed PSDM shown in Equation (3), it is significant to evaluate the effectiveness of an IM before the efficiency assessment. Effectiveness of a specific IM can be defined by its coefficient of determination ($R^2$) of the predicted PSDM, a value between 0 and 1 [12]. A bigger value of $R^2$ strongly demonstrates the effectiveness of a given IM. On the other hand, if the value of $R^2$ is too small, the IM is not effective enough, and evaluation of other required criteria will not need to be further conducted [12].

### 3.2. Practicality

Practicality can indicate the dependency of the EDP on the investigated IM. For the linear PSDM as given in Equation (4), this criterion can be quantified by using the parameter $b$ in Equation (4). A value of $b$ close to zero indicates that the IM contributes less significantly to predicting the structural demand, indicating an impractical IM. However, a higher value of $b$ suggests a strong dependency between the seismic IM and the structural demand [2].

### 3.3. Proficiency

By considering the composite contributions of practicality and efficiency, Padgett et al. [10] suggested proficiency as a combined criterion of practicality and efficiency, which can be represented by using Equation (5). This evaluation criterion is also known as the modified dispersion $\xi_\beta$, which is calculated by Equation (6). A lower value of $\xi_\beta$ contributes to a more proficient IM, indicating a stronger correlation relationship of the seismic IM and the structural demand while less dispersion around the median values of the PSDMs [2].

$$P(EDP \geq D | IM) = \Phi \left( \frac{\ln(IM) - \frac{\ln(D) - \ln(a)}{b}}{\frac{\beta_{D/IM}}{b}} \right) \tag{5}$$

$$\xi_\beta = \frac{\beta_{D/IM}}{b} \tag{6}$$

### 3.4. Sufficiency

Sufficiency reflects the dependency of the investigated IM on the seismic parameters, such as magnitude ($M$) and source-to-site distance ($R_d$). Based on many previous studies [12,26–28], a sufficient IM should be conditionally statistically independent of the $M$ and $R_d$. Sufficiency of a given IM can be determined by performing a regression analysis on the residuals between the actual structural response and the predicted PSDM that is related to the $M$ or $R_d$. Then, the *p*-value from the regression analysis of the residuals can be employed to determine the IM sufficiency [2,12,28], which suggests the probability of rejecting the null hypothesis that the slope coefficient of linear regression is zero. The present study employs a 5% significance level ($p = 0.05$) as the threshold for the IM sufficiency evaluation. Thus, the seismic IM which leads to a *p*-value less than this threshold will be considered to be insufficient [12].

### 3.5. Hazard Computability

Although a given IM may be considered as suitable with respect to these above-mentioned metrics, a specific IM may be less desirable because of a lack of the required seismic hazard models [12]. In this regard, Giovernale et al. [8] suggested the hazard computability of a given IM as a measure to determine the required endeavor to perform the probabilistic seismic hazard analysis or construct the seismic hazard curve, $\lambda(im)$ [12]. It can be concluded that the hazard computability of the input seismic records is critical in determining an appropriate IM. This was discussed in many existing studies [9,12,23,29,30].

## 4. Case Study: FE Modeling and Engineering Demand Parameters

### 4.1. Bridge Description and FE Modeling

This study takes a representative multi-span reinforced concrete (RC) continuous girder (MSRCCG) bridge as the case study, which has five spans, 30 m each, and a 16 m wide superstructure supported by four RC circular piers and two RC abutments. The superstructure consists of a 1.8 m high box girder and a cap beam. The height of each pier is 10 m. Detailed geometric information of the bridge is shown in Figure 2. According to the design guidelines given in [31], the reinforcing ratios of the longitudinal steel bars and transverse spiral hoops are 1.08% and 0.58%, respectively. Structural loads can be transferred to the abutments and piers through the plate-type elastomeric bearing (PTEB) and the lead rubber bearing (LRB), respectively. Nine RC piles 30 m long and with a diameter of 1.5 m are arranged for the pier foundation system, and the soil conditions of the bridge are considered as medium-hard soil.

Although information regarding the finite element (FE) modeling of the bridge can be found in the authors' previous studies [3–7], a detailed description of the numerical modeling of the case study bridge is also provided herein. The three-dimensional non-linear FE model of the bridge as shown in Figure 2 is developed by using the OpenSEES program [32] to simulate the seismic response of the bridge. For example, the composite action of the deck and cap beam is modeled using the linear elastic beam–column elements since their damage is not expected in the bridge superstructure during earthquake events. Bridge piers are modeled using nonlinear beam–column elements with fiber defined cross-sections considering the axial force–moment interaction and material nonlinearities. For the fiber-element model of RC piers, the stress–strain relationship of the confined and unconfined concrete is modeled as Concrete 04 material, whereas the longitudinal steel bars, as well as the transverse spiral hoops, are simulated using the Steel 02 material, both of which are available material models in the OpenSEES database [32]. In addition, linear

translational and rotational springs are utilized to simulate the pile foundations under the piers to capture the translation and rotation behaviors of the foundation system. The stiffness of these springs is determined by the "m" method according to the guidelines for the seismic design of Chinese highway bridges [31]. Moreover, the PTEB and LRB bearings are simulated by using the elastomeric bearing (plasticity) element, and the behavior of abutments is considered by incorporating the contribution of back-fill soil and piles, which can be modeled by using the hyperbolic material and the hysteretic material available in the OpenSEES database [32], respectively. The transverse concrete stoppers are simulated by the hysteretic material and elastic–perfectly plastic gap elements. The pounding effect between the deck and abutments can be simulated using the contact element (i.e., non-linear translational springs) considering the effects of hysteretic energy loss, which can be simulated by impact materials in the OpenSEES database [32]. The three-dimensional nonlinear dynamic FE model of the case study bridge and the force–deformation backbone curves of all critical bridge components are summarized in Figure 2, and the corresponding parameters indicating the nonlinearities involved in the boundary conditions are given in Table 1.

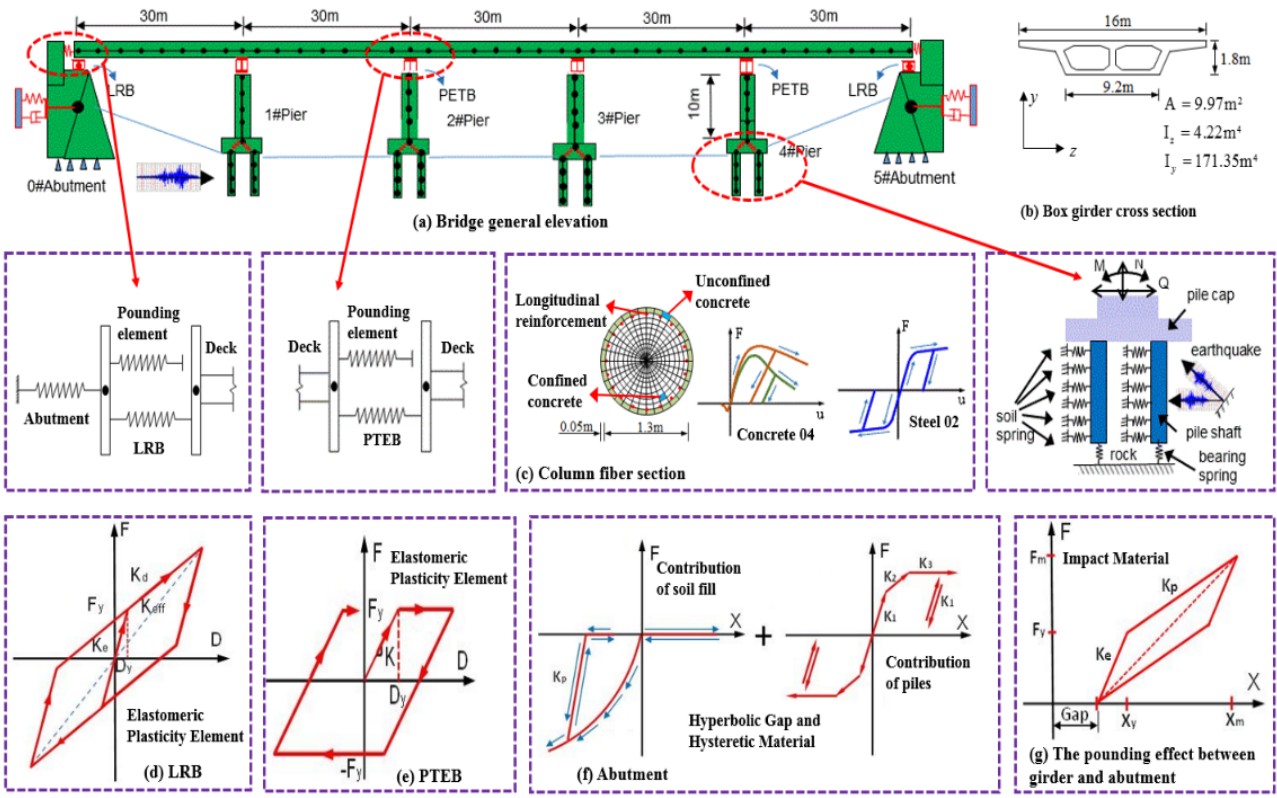

**Figure 2.** Schematic nonlinear dynamic modeling of the case study bridge.

Uncertainties are inherently involved in the input ground motions, material, structural geometry, and boundary conditions. The authors previously proposed a schematic seismic vulnerability assessment framework for highway bridges in consideration of many modeling-related uncertain parameters, including (i) the structurally related uncertainty (SU) parameters, (ii) material-related uncertainty (MU) parameters, and (iii) boundary condition-related uncertainty (BU) parameters [5,7]. However, since the present study is trying to investigate the selection of the ground motion IMs and uncertainty evaluation of ground motions in the PSDA of highway bridges, all modeling-related uncertain parameters are equal to their respective median values (deterministic), which are summarized in Table 1. Detailed information regarding the introductions of these modeling-related uncertain parameters can be found in [5,7].

**Table 1.** Summary of the modeling-related parameters of the case study bridge.

| | Parameters | Description | Value | Units |
|---|---|---|---|---|
| Structure-related parameters | $\lambda_w$ | Concrete weight coefficient | 1.04 | — |
| | $D$ | Pier diameter | 1.4 | m |
| | $c$ | Concrete cover thickness | 0.05 | m |
| | $\varphi$ | Longitudinal reinforcement diameter | 28 | mm |
| | $\xi$ | Damping ratio | 0.05 | — |
| Material-related parameters | $E_c$ | Young's modulus of concrete | $3 \times 10^4$ | MPa |
| | $f_{c,\,cover}$ | The peak strength of cover concrete | 27.58 | MPa |
| | $\varepsilon_{c,cover}$ | Peak strain of cover concrete | 0.002 | — |
| | $\varepsilon_{cu,cover}$ | The ultimate strain of cover concrete | 0.006 | — |
| | $f_{c,\,core}$ | The peak strength of core concrete | 34.47 | MPa |
| | $\varepsilon_{c,core}$ | Peak strain of core concrete | 0.005 | 0.005 |
| | $\varepsilon_{cu,core}$ | The ultimate strain of core concrete | 0.02 | 0.02 |
| | $E_s$ | Young's modulus of steel rebar | $2 \times 10^5$ | MPa |
| | $f_y$ | Yield strength of steel rebar | 335 | MPa |
| | $\gamma$ | Post-yield to initial stiffness ratio | 0.02 | — |
| Boundary condition-related parameters | $\mu_{PETB}$ | The friction coefficient of PTEB | 0.15 | — |
| | $G_{PETB}$ | Shear modulus of PTEB | 1180 | MPa |
| | $K_{P\_LRB}$ | Post-yield stiffness of LRB | 1500 | kN/m |
| | $P_{ult}$ | Abutment ultimate capacity | 10,853 | kN |
| | $K_{passive}$ | Abutment passive stiffness | $3.04 \times 10^5$ | kN/m |
| | $K_{active}$ | Abutment active stiffness | $1.86 \times 10^4$ | kN/m |
| | $K_{eff}$ | Pounding effective stiffness | $1.94 \times 10^6$ | kN/m |

*4.2. Engineering Demand Parameters (EDPs)*

It is significant to define the bridge EDPs to obtain the peak structural responses from the NTHAs. Several critical bridge EDPs are defined as given in Table 2, including the critical responses referring to the curvature ductility at the base section of the piers ($u_\Phi$), the relative displacement of LRB ($\delta_{LRB}$), the relative displacement of PTEB ($\delta_{PTEB}$), the active and passive deformations of the abutments ($\Delta_{Abut\_active}$ and $\Delta_{Abut\_passive}$).

**Table 2.** The considered bridge engineering demand parameters (EDPs).

| ID | EDP | Abbreviation | Unit | Note |
|---|---|---|---|---|
| 1 | Curvature ductility of the pier | $u_{\Phi\_L}$ | $m^{-1}$ | Longitudinal |
| 2 | The curvature ductility of the pier | $u_{\Phi\_T}$ | $m^{-1}$ | Transverse |
| 3 | Relative displacement of the LRB | $\delta_{LRB\_L}$ | cm | Longitudinal |
| 4 | Relative displacement of the LRB | $\delta_{LRB\_T}$ | cm | Transverse |
| 5 | Relative displacement of the PTEB | $\delta_{PTEB\_L}$ | cm | Longitudinal |
| 6 | Relative displacement of the PTEB | $\delta_{PTEB\_T}$ | cm | Transverse |
| 7 | Abutment deformation | $\Delta_{Abut\_active}$ | cm | Active |
| 8 | Abutment deformation | $\Delta_{Abut\_passive}$ | cm | Passive |

## 5. Ground Motion Records and the Considered Ground Motion IMs

A ground motion bin approach can be applied to perform the PSDA of highway bridges [2,33]. Two bins with 100 near-field and 100 far-field ground motions are selected from the PEER Strong Motion Database [34], respectively. Figure 3 gives the information of the selected ground motions, including the plots of the $M–R_d$ relation and the distribution of *PGV* values. Figure 4 shows the response spectra (in terms of the *SA* under the Rayleigh damping ratio of 5%) of the selected near-field and far-field ground motions, respectively. Moreover, comparison of the median response spectra of the selected near-field and far-field earthquake records is shown in Figure 5. As seen from Figure 5, in the short period phase (i.e., 0 < T < 0.7 s), the median spectra of these two ground motion bins are almost the

same, while the *SA* of the near-field ground motions is significantly greater than that of their far-field counterparts, indicating the pulse-like effect of the near-field seismic records. Furthermore, in the present study, twenty-eight IM candidates as given in Table 3 are chosen from the previous studies [11,19,24] and they are examined for the PSDA of the bridge. As shown in Table 3, these ground motion IMs can be categorized as the displacement-related, velocity-related, acceleration-related, and time-related IMs, respectively.

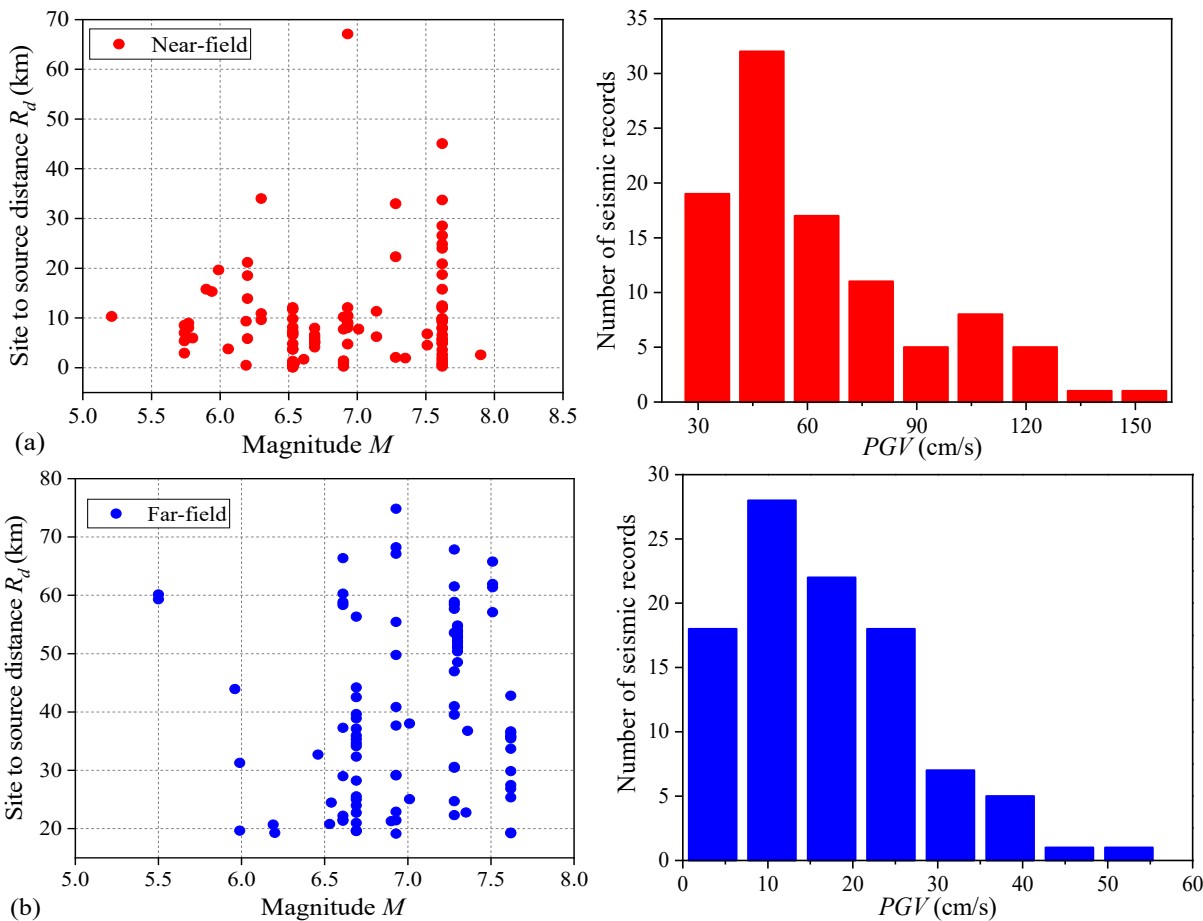

**Figure 3.** Information of the selected ground motions: (**a**) the near-field and (**b**) far-field ground motions.

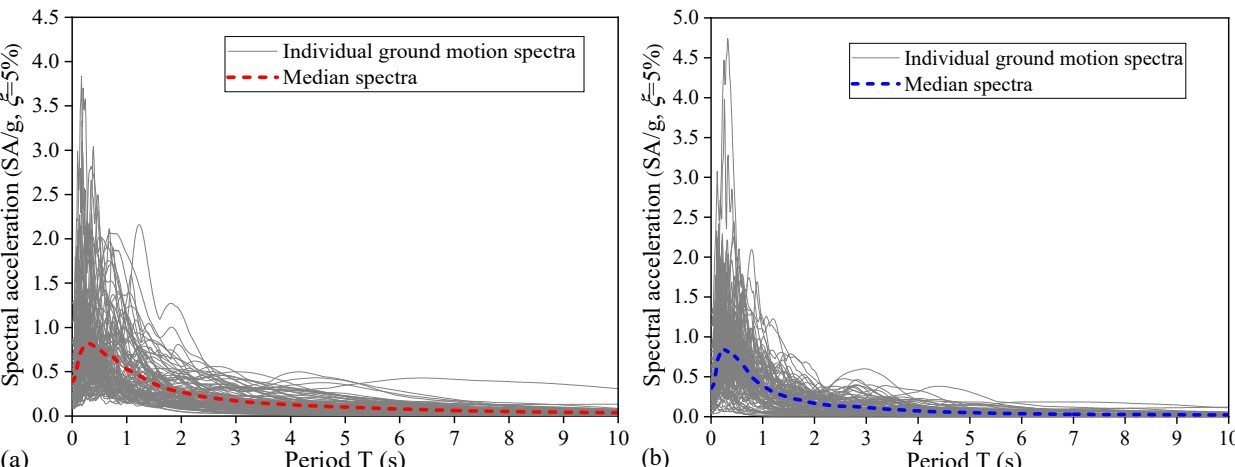

**Figure 4.** Response spectra of the selected ground motions after scaling: (**a**) the near-field; (**b**) far-field ground motions.

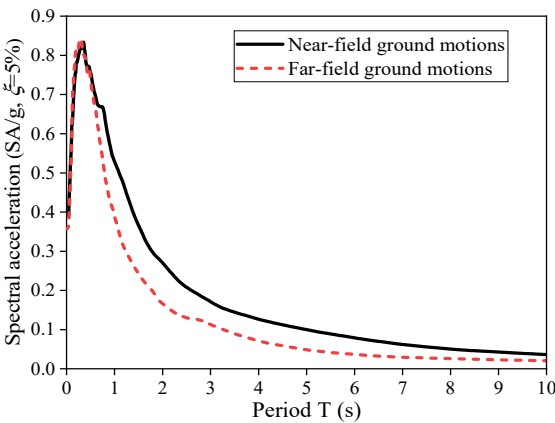

**Figure 5.** Comparison of the median response spectra of the selected ground motions after scaling.

**Table 3.** Summary of the considered ground motion IMs.

| IM Number | IM Name | Definition | Calculation Method | Unit |
|:---:|:---:|:---:|:---:|:---:|
| 1 | $PGD$ | Peak ground displacement | $\max\lvert u_g(t)\rvert$ | cm |
| 2 | $PGV$ | Peak ground velocity | $\max\lvert \dot{u}_g(t)\rvert$ | cm/s |
| 3 | $PGA$ | Peak ground acceleration | $\max\lvert \ddot{u}_g(t)\rvert$ | g |
| 4 | $SD$ | Spectra displacement | $SD(T) = \frac{T}{2\pi}\lvert \int_0^T \ddot{u}_g(t)e^{-\xi\omega(t-\tau)}\sin\omega(t-\tau)d\tau\rvert_{\max}$ | cm |
| 5 | $SV$ | Spectral velocity | $SV(T) = \frac{2\pi}{T}SV(T)^2$ | cm/s |
| 6 | $SA$ | Spectral acceleration | $SA(T) = \left(\frac{2\pi}{T}\right)SD(T)$ | g |
| 7 | $CSA$ | Cordova spectral acceleration | $CSA(T,\xi) = SA(T1,\xi)\left[\frac{SA(cT1,\xi)}{SA(T1,\xi)}\right]^\alpha$ | cm/s² |
| 8 | $EPD$ | Effective peak displacement | $EPD = \frac{SD_{avg}(T_i,\xi)\big\vert_{2.5}^{T_i=4.0}}{2.5}$ | cm |
| 9 | $EPV$ | Effective peak velocity | $EPV = \frac{SV_{avg}(T_i,\xi)\big\vert_{0.8}^{T_i=2.0}}{2.5}$ | cm/s |
| 10 | $EPA$ | Effective peak acceleration | $EPA = \frac{SA_{avg}(T_i,\xi)\big\vert_{0.1}^{T_i=0.5}}{2.5}$ | cm/s² |
| 11 | $DSI$ | Displacement response intensity | $DSI = \int_{0.7}^{2.0} SV(T,\xi=0.05)dT$ | cm |
| 12 | $VSI$ | Displacement velocity intensity | $VSI = \int_{0.7}^{2.0} SV(T,\xi=0.05)dT$ | cm/s |
| 13 | $ASI$ | Acceleration velocity intensity | $ASI = \int_{0.1}^{0.5} SA(T,\xi=0.05)dT$ | g |
| 14 | $SI$ | Response spectrum intensity | $SI = \int_{0.1}^{2.5} SV(T,\xi=0.05)dT$ | cm |
| 15 | $I_A$ | Arias intensity | $I_A = \frac{\pi}{2g}\int_0^{D_f}\left[\ddot{u}_g(t)\right]^2 dt$ | cm/s |
| 16 | $T_D$ | Strong motion duration | $T_D = t(0.95I_A) - t(0.05I_A)$ | s |
| 17 | $D_{rms}$ | Root mean square displacement | $D_{rms} = \sqrt{\frac{1}{T_D}\int_0^{D_f}\left[u_g(t)\right]^2 dt}$ | cm |
| 18 | $V_{rms}$ | Root mean square velocity | $V_{rms} = \sqrt{\frac{1}{T_D}\int_0^{D_f}\left[\dot{u}_g(t)\right]^2 dt}$ | cm/s |
| 19 | $A_{rms}$ | Root mean square acceleration | $A_{rms} = \sqrt{\frac{1}{T_D}\int_0^{D_f}\left[\ddot{u}_g(t)\right]^2 dt}$ | cm/s² |
| 20 | $CAI$ | Cumulative absolute impulse | $CAI = \int_0^{D_f}\lvert u_g(t)\rvert dt$ | cm-s |
| 21 | $CAD$ | Cumulative absolute displacement | $CAD = \int_0^{D_f}\lvert \dot{u}_g(t)\rvert dt$ | cm |
| 22 | $CAV$ | Cumulative absolute velocity | $CAV = \int_0^{D_f}\lvert \ddot{u}_g(t)\rvert dt$ | cm/s |
| 23 | $I_m$ | Median period intensity measure | $I_M = PGV(T_D^{0.25})$ | cm/s^0.75 |
| 24 | $I_C$ | Characteristic intensity | $I_C = A_{rms}^{1.5}T_D^{0.5}$ | cm^1.5/s^2.5 |

**Table 3.** *Cont.*

| IM Number | IM Name | Definition | Calculation Method | Unit |
|---|---|---|---|---|
| 25 | $I_D$ | Displacement intensity | $I_D = \frac{1}{PGD} \int_0^{D_f} \left[ u_g(t) \right]^2 dt$ | cm-s |
| 26 | $I_V$ | Velocity intensity | $I_V = \frac{1}{PGV} \int_0^{D_f} \left[ \dot{u}_g(t) \right]^2 dt$ | cm |
| 27 | FR1 | Frequency ratio 1 | $FR1 = PGV/PGA$ | s |
| 28 | FR2 | Frequency ratio 2 | $FR2 = PGD/PGV$ | s |

## 6. Results and Discussions of the Selection of Ground Motion IMs

### 6.1. Selection of IMs for the Near-Field Ground Motions

Figure 6 shows the effectiveness evaluation of the considered IMs for different EDPs of the bridge listed in Table 2. As shown in Figure 6, for the near-field ground motions, coefficients of determination ($R^2$) of the predicted PSDMs for different bridge components are all less than 0.9, especially those displacement-related IMs such as *PGD*, *EPD*, and *DSI*, and those time-related IMs except $I_A$ and $A_{rms}$, as well as those hybrid IMs except $I_C$, which are less than 0.2. According the criterion of effectiveness evaluation of IM introduced in Section 3, these IMs that contribute to a very small value of $R^2$ will be considered as less effective and they will not be further evaluated for other criteria. By ignoring these ineffective IMs, Figure 7 shows the proficiency and efficiency evaluation results of these 14 remaining seismic IMs.

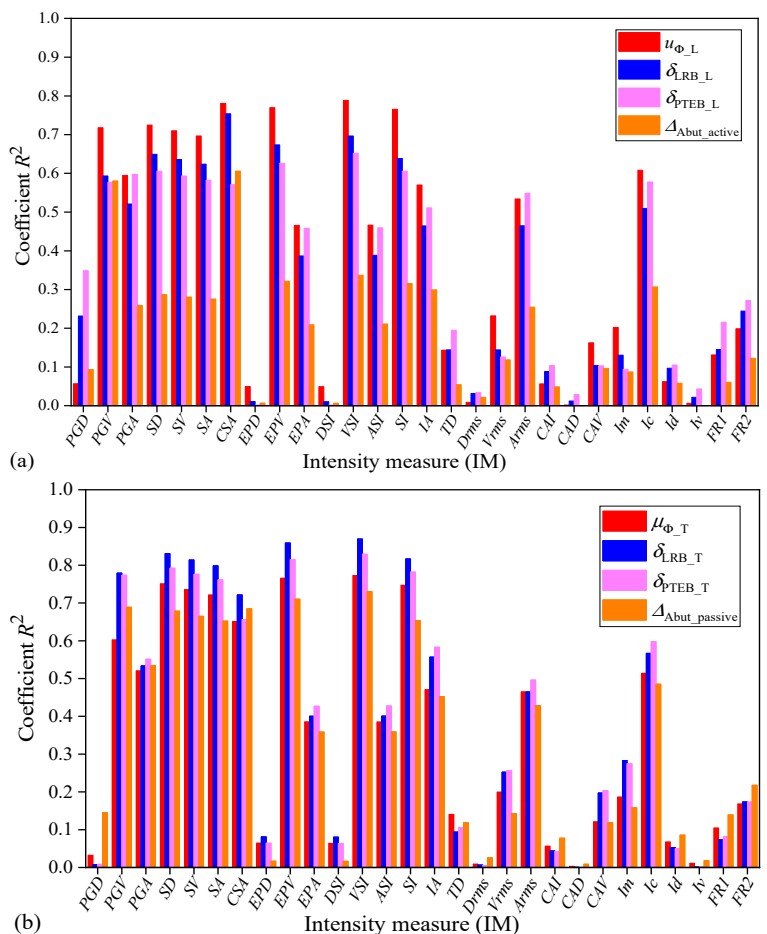

**Figure 6.** IM effectiveness for different bridge EDPs under near-field ground motions. (**a**) longitudinal and active; (**b**) transverse and passive.

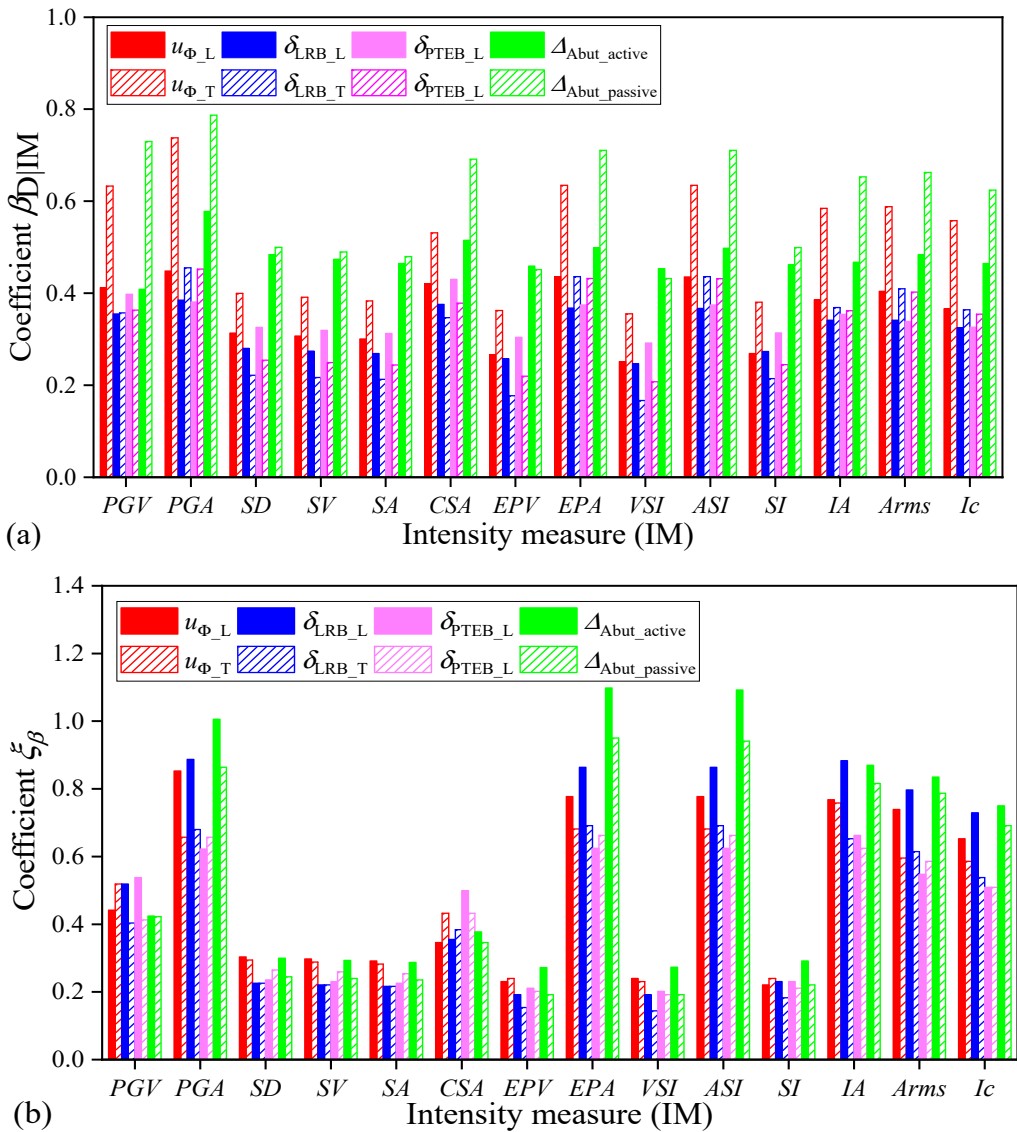

**Figure 7.** IM efficiency and proficiency evaluation for different EDPs under the near-field ground motions: (**a**) efficiency evaluation and (**b**) proficiency evaluation.

As observed from Figure 7a, for these 14 IMs after the effectiveness evaluation, they tend to generate relatively good predicted PSDMs. This indicates an effective IM (with higher value of $R^2$) may lead to an efficient IM (quantified in $\beta_{D|IM}$). Moreover, values of $\beta_{D|IM}$ of the developed PSDMs for different bridge components are less than 0.5, except those for the bridge EDPs such as $u_{\Phi\_T}$ and $\Delta_{Abut\_passive}$. Hence, these 14 IMs can be considered to be efficient IMs in terms of $\beta_{D|IM}$, so they need to be further examined in proficiency and sufficiency evaluation. Then, based on the composite measure of efficiency and practicality, Figure 7b shows the proficiency evaluation results of these IMs. As seen from Figure 7, after considering the criterion of practicality, variation in proficiency evaluation of these 14 efficient IMs is significant. Based on the indicator of $\xi_\beta$, these spectrum-related IMs (*SA*, *SV*, and *SD*) and velocity-related ground motion IMs (*EPV*, *VSI*, and *SI*) tend to have good proficiency, whereas the acceleration-rated IMs, such as *PGA*, *EPA*, *ASI*, $A_{rms}$, $I_A$, and $I_C$, are relatively not proficient. In addition, Figure 8 shows the top ten proficient IMs for different bridge components. With such a plot, bridge owners can easily select more proficient IMs for the considered bridge components for the PSDA of highway bridges in future.

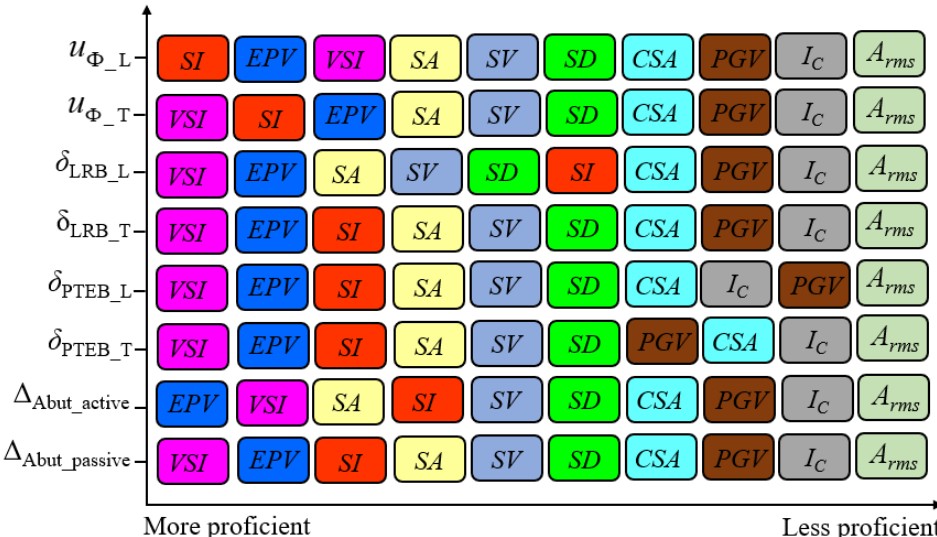

**Figure 8.** Top ten proficient IMs for different bridge EDPs under the near-field ground motions.

Moreover, as seen from the proficiency evaluation of the considered IMs in Figure 7b, values of $\xi_\beta$ for the spectrum-related IMs such as *SA*, *SV*, and *SD* are almost the same, and *SA* is the representative candidate IM among these three IMs to investigate the seismic performance and seismic vulnerability of highway bridges by many previous studies [3–7]. Thus, in the following sufficiency evaluation of the considered IMs, these proficient IMs, including *PGV*, *SA*, *CSA*, *EPV*, *VSI*, and *SI*, are investigated. Tables 4 and 5 show the *p*-values for the residuals of the developed PSDMs for different bridge components and the *M* and $R_d$, respectively, for the near-field ground motions. As shown in Table 4, almost all these considered IMs do not satisfy the sufficiency requirement for the magnitude *M*. In specific, among the considered IMs in sufficiency evaluation, *PGV* and *CSA* are the least sufficient IMs for all bridge EDPs. Furthermore, *SA*, *EPV*, and *VSI* tend to have good sufficiency for the structural response of abutment ($\Delta_{Abut\_active}$ and $\Delta_{Abut\_passive}$) and pier ($u_{\Phi\_L}$), whereas they are not sufficient for other EDPs, such as $u_{\Phi\_T}$ and $\delta_{LRB\_L}$. Moreover, *SI* seems have the best sufficiency for all EDPs among these considered IMs for M of the near-field ground motions. However, as seen from Table 5, compared to the sufficiency evaluation of the considered IMs for *M* as given in Table 4, it shows a different trend for the sufficiency evaluation for $R_d$. All IMs except *CSA* satisfy the sufficiency requirement for $R_d$ for the near-field ground motions. Among these considered IMs, the most sufficient IM varies for different bridge EDPs. For example, *SI* is the best IM candidate to investigate the seismic response of abutments ($\Delta_{Abut\_active}$ and $\Delta_{Abut\_passive}$), whereas *SA* is the most proficient IM for the piers ($u_{\Phi\_L}$ and $u_{\Phi\_T}$).

**Table 4.** IM sufficiency evaluation in terms of *M* for near-field ground motions.

| IM | $u_{\Phi\_L}$ | $u_{\Phi\_T}$ | $\delta_{LRB\_L}$ | $\delta_{LRB\_T}$ | $\delta_{PTEB\_L}$ | $\delta_{PTEB\_T}$ | $\Delta_{Abut\_active}$ | $\Delta_{Abut\_passive}$ |
|---|---|---|---|---|---|---|---|---|
| *PGV* | 0.000 | 0.000 | 0.000 | 0.000 | 0.000 | 0.000 | 0.048 | 0.000 |
| *SA* | 0.182 | 0.000 | 0.038 | 0.038 | 0.000 | 0.067 | 0.490 | 0.806 |
| *CSA* | 0.000 | 0.000 | 0.000 | 0.000 | 0.000 | 0.000 | 0.048 | 0.000 |
| *EPV* | 0.077 | 0.000 | 0.010 | 0.000 | 0.000 | 0.019 | **0.614** | **0.883** |
| *VSI* | 0.173 | 0.000 | 0.038 | **0.106** | 0.000 | 0.048 | 0.442 | 0.672 |
| *SI* | **0.288** | 0.005 | **0.154** | 0.096 | **0.086** | **0.144** | 0.403 | 0.538 |

Note: Value in bold in each column in the table indicates the most sufficient IM for each EDP.

**Table 5.** IM sufficiency evaluation in terms of $R_d$ for near-field ground motions.

| IM | $u_{\Phi\_L}$ | $u_{\Phi\_T}$ | $\delta_{LRB\_L}$ | $\delta_{LRB\_T}$ | $\delta_{PTEB\_L}$ | $\delta_{PTEB\_T}$ | $\Delta_{Abut\_active}$ | $\Delta_{Abut\_passive}$ |
|---|---|---|---|---|---|---|---|---|
| *PGV* | 0.038 | 0.125 | 0.067 | 0.086 | 0.125 | 0.096 | 0.566 | 0.566 |
| *SA* | **0.269** | **0.230** | **0.490** | **0.365** | 0.816 | **0.346** | 0.528 | 0.326 |
| *CSA* | 0.048 | 0.019 | 0.058 | 0.038 | 0.096 | 0.038 | 0.634 | 0.614 |
| *EPV* | **0.269** | 0.202 | 0.422 | 0.336 | **0.912** | 0.326 | 0.595 | 0.269 |
| *VSI* | 0.250 | 0.192 | 0.384 | 0.317 | 0.893 | 0.326 | 0.634 | 0.307 |
| *SI* | 0.173 | 0.125 | 0.211 | 0.182 | 0.614 | 0.192 | **0.950** | **0.682** |

Note: Value in bold in each column in the table indicates the most sufficient IM for each EDP.

*6.2. Selection of IMs for the Far-Field Ground Motions*

The IM selection procedure is also investigated for the far-field ground motions. First, according to the criterion of effectiveness qualification, those considered IMs with too small values of $R^2$ of the predicted PSDMs for different bridge components, such as *PGA*, *PGD*, *EPA*, *ASI*, and *FR*1, are less effective seismic IMs, so they will not need to be further considered for the efficiency evaluation. Thus, Figure 9 shows the proficiency and efficiency evaluation results of the 18 remaining seismic IMs for the far-field ground motions. As observed in Figure 9, the variation in proficiency evaluation of these efficient IMs is significant. Based on the indicator of $\xi_\beta$, these spectrum-related IMs (*SA*, *SV*, *SD*, and *CSA*) and velocity-related IMs (*PGV*, *EPV*, *VSI*, *SI*, and $V_{rms}$) tend to have good efficiency and proficiency. In addition, as seen from Figure 9b, values of $\xi_\beta$ for the spectrum-related IMs such as *SA*, *SV*, and *SD* are almost the same, thus, in the following sufficiency evaluation of the considered IMs, these proficient IMs, including *PGV*, *SA*, *CSA*, *EPV*, *VSI*, *SI*, $V_{rms}$, and $I_m$, are investigated in the following.

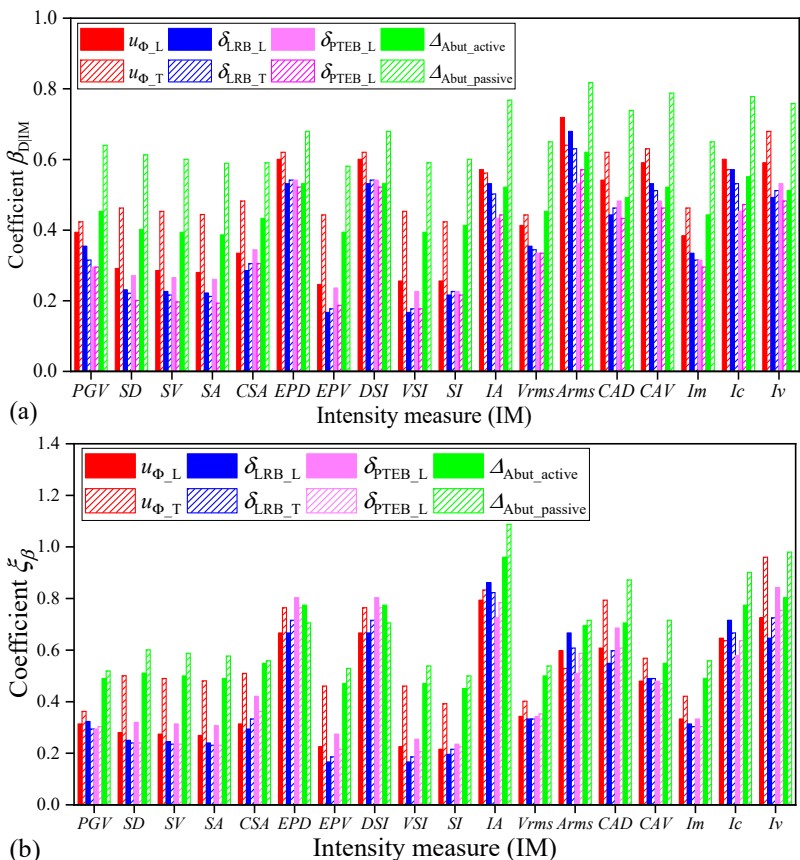

**Figure 9.** IM efficiency and proficiency evaluation for different EDPs under the far-field ground motions: (**a**) efficiency evaluation and (**b**) proficiency evaluation.

Following the sufficiency evaluation procedure, Tables 6 and 7 show the *p*-values for the residuals of the developed PSDMs for different bridge components and the *M* and $R_d$ for the far-field ground motions, respectively. As seen from Tables 6 and 7, all these considered IMs satisfy the sufficiency requirement for *M* and $R_d$. Among these consider IMs, the most sufficient IM varies for different bridge EDPs. For instance, based on the sufficiency evaluation in terms of *M*, *PGV* is the best IM candidate for $\Delta_{\text{Abut\_active}}$, whereas that for $\Delta_{\text{Abut\_passive}}$ is *SA*. Likewise, *SA* is the best IM candidate for $u_{\Phi\_L}$, while that for $u_{\Phi\_T}$ is $I_m$. In addition, Figure 10 shows the top ten proficient IMs for different bridge EDPs under the far-field ground motions, which can help bridge owners select more proficient IMs for the PSDA of highway bridges. In summary, based on the above introduction of the ground motion IM selection for both the near-field and far-field ground motions, several findings are summarized in Table 8.

**Table 6.** IM sufficiency evaluation in terms of *M* for the far-field ground motions.

| IM | $u_{\Phi\_L}$ | $u_{\Phi\_T}$ | $\delta_{\text{LRB\_L}}$ | $\delta_{\text{LRB\_T}}$ | $\delta_{\text{PTEB\_L}}$ | $\delta_{\text{PTEB\_T}}$ | $\Delta_{\text{Abut\_active}}$ | $\Delta_{\text{Abut\_passive}}$ |
|---|---|---|---|---|---|---|---|---|
| *PGV* | 0.571 | 0.483 | 0.699 | 0.798 | 0.808 | 0.837 | **0.699** | 0.798 |
| *SA* | **0.955** | 0.483 | 0.680 | 0.630 | 0.571 | 0.561 | 0.335 | **0.973** |
| *CSA* | 0.660 | 0.690 | 0.424 | 0.552 | 0.601 | 0.630 | 0.315 | 0.719 |
| *EPV* | 0.808 | 0.384 | 0.660 | 0.739 | 0.640 | 0.749 | 0.384 | 0.965 |
| *VSI* | 0.640 | 0.325 | **0.867** | **0.916** | 0.729 | **0.906** | 0.443 | 0.887 |
| *SI* | 0.335 | 0.236 | 0.532 | 0.532 | 0.926 | 0.581 | 0.680 | 0.719 |
| $V_{rms}$ | 0.512 | 0.414 | 0.670 | 0.670 | **0.955** | 0.670 | 0.670 | 0.739 |
| $I_m$ | 0.532 | **0.916** | 0.355 | 0.315 | 0.217 | 0.286 | 0.217 | 0.798 |

Note: Value in bold in each column in the table indicates the most sufficient IM for each EDP.

**Table 7.** IM sufficiency evaluation in terms of $R_d$ for the far-field ground motions.

| IM | $u_{\Phi\_L}$ | $u_{\Phi\_T}$ | $\delta_{\text{LRB\_L}}$ | $\delta_{\text{LRB\_T}}$ | $\delta_{\text{PTEB\_L}}$ | $\delta_{\text{PTEB\_T}}$ | $\Delta_{\text{Abut\_active}}$ | $\Delta_{\text{Abut\_passive}}$ |
|---|---|---|---|---|---|---|---|---|
| *PGV* | 0.176 | 0.598 | 0.323 | **0.941** | **0.862** | **0.794** | 0.647 | 0.539 |
| *SA* | **0.951** | 0.372 | 0.588 | 0.137 | 0.108 | 0.167 | 0.686 | **0.921** |
| *CSA* | 0.637 | 0.059 | 0.372 | 0.049 | 0.108 | 0.098 | 0.539 | 0.804 |
| *EPV* | 0.412 | 0.176 | 0.627 | 0.147 | 0.294 | 0.294 | **0.976** | 0.804 |
| *VSI* | 0.363 | 0.196 | 0.539 | 0.176 | 0.323 | 0.333 | 0.970 | 0.794 |
| *SI* | 0.137 | 0.314 | 0.284 | 0.578 | 0.657 | 0.764 | 0.804 | 0.657 |
| $V_{rms}$ | 0.127 | **0.764** | 0.206 | 0.764 | 0.715 | 0.676 | 0.529 | 0.470 |
| $I_m$ | 0.588 | 0.235 | **0.813** | 0.402 | 0.451 | 0.519 | 0.970 | 0.843 |

Note: Value in bold in each column in the table indicates the most sufficient IM for each EDP.

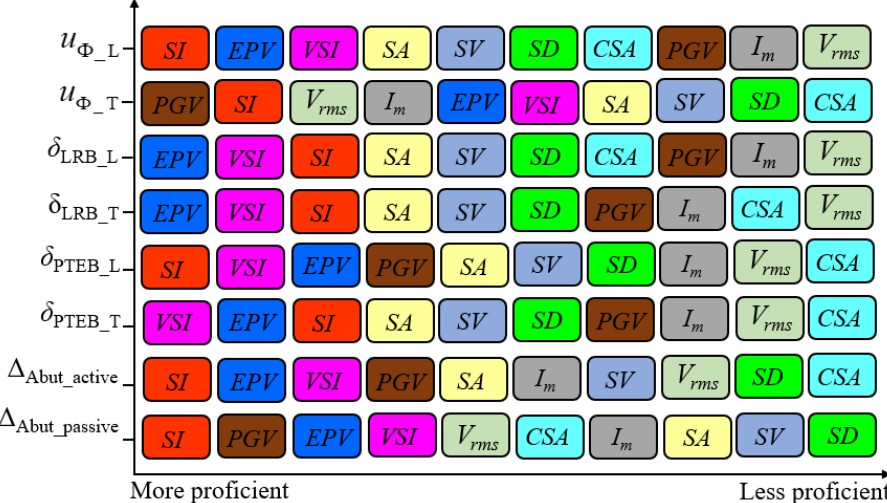

**Figure 10.** Top ten proficient IMs for different bridge EDPs under the far-field ground motions.

**Table 8.** Summary of evaluation of different IMs for both the near-field and far-field ground motions.

| Ground Motions | (i) Efficiency; (ii) Practicality; and (iii) Proficiency Evaluation | (iv) Sufficiency Evaluation | |
|---|---|---|---|
| | | **Magnitude (*M*)** | **Source-to-Site Distance ($R_d$)** |
| Near-field | *SA*, *SV*, *SD*, *CSA*, *PGV*, *EPV*, *VSI*, and *SI*. | (1) The optimal IM varies for different bridge EDPs. (2) All of the eight considered IMs do not satisfy sufficiency requirement for *M*. | (1) The optimal IM varies for different bridge EDPs. (2) All of the eight considered IMs except *CSA* satisfy the sufficiency requirement for $R_d$. |
| Far-field | *SA*, *SV*, *SD*, *CSA*, *PGV*, *EPV*, *VSI*, *SI*, $V_{rms}$, and $I_m$. | (1) The optimal IM varies for different bridge EDPs. (2) All of the ten considered IMs satisfy the sufficiency requirement for *M*. | |

As seen from Table 8, during the evaluation of the seismic IMs, there are many candidate IMs that can satisfy the requirements of efficiency, practicality, and proficiency. However, after further performing the sufficiency evaluation, the number of the available sufficient IMs will decrease significantly. Thus, from the above discussions, in the PSDA of highway bridges, it is critical to select the appropriate seismic IMs to develop the predicted PSDMs for different bridge components to further investigate their seismic responses.

## 7. Effect of Uncertainties in Ground Motions on the PSDA of Highway Bridges

Since the *PGA* is one of the most widely employed seismic IMs in the seismic fragility analysis of highway bridges, by taking the *PGA* as the seismic IM and after a series of NTHAs are conducted, Figure 11 shows comparison of the developed PSDMs for different bridge components under both the near-field and far-field ground motions, respectively. As seen from Figure 11, for a given bridge component, the developed PSDMs for $u_{\Phi\_L}$ and $u_{\Phi\_T}$ are significantly different for the piers under the near-field and far-field ground motions. This can be attributed to the pulse-like effect of near-field ground motions. The pulse-like effect will lead to higher seismic responses and more severe destruction for the bridge structures, so more attention should be paid to the PSDA of highway bridges under the pulse-like near-fault ground motions.

Moreover, as observed from Figure 11, the dispersions of the obtained PSDMs for different bridge EDPs are a bit larger compared to the previous studies in the literature. This may have resulted from the developed FE model of the case study bridge considering various kinds of nonlinear dynamic effects. For example, as seen from Figure 11d, compared to other bridge components, the dispersions of the developed PSDMs for the abutments are relatively larger. This may because of the nonlinear pounding effects (i.e., complicated nonlinearities involved in the consideration of boundary conditions) of the girder and abutments are taken into consideration in the modeling of the bridge. Additionally, as seen from Figure 11, the developed PSDMs under different ground motion bins are significantly different. This suggests that uncertainty stemming from the BTB variability of seismic records may lead to a great difference in the obtained PSDMs, and it also suggests the importance of the selection of earthquake records in the PSDA, the following seismic risk, and seismic fragility analyses of highway bridges. In addition, for a given selected ground motion bin, uncertainty stemming from the RTR variability of the input seismic records in this bin (i.e., spectrum-related characteristics are different) may also contribute to the dispersions of the developed PSDMs. Such a dispersion of a PSDM derived from the RTR variability of ground motion can be quantified by the logarithmic standard deviation $\beta_{D|IM}$ in Equation (4), and the greater $\beta_{D|IM}$, the greater the dispersion of the developed PSDMs.

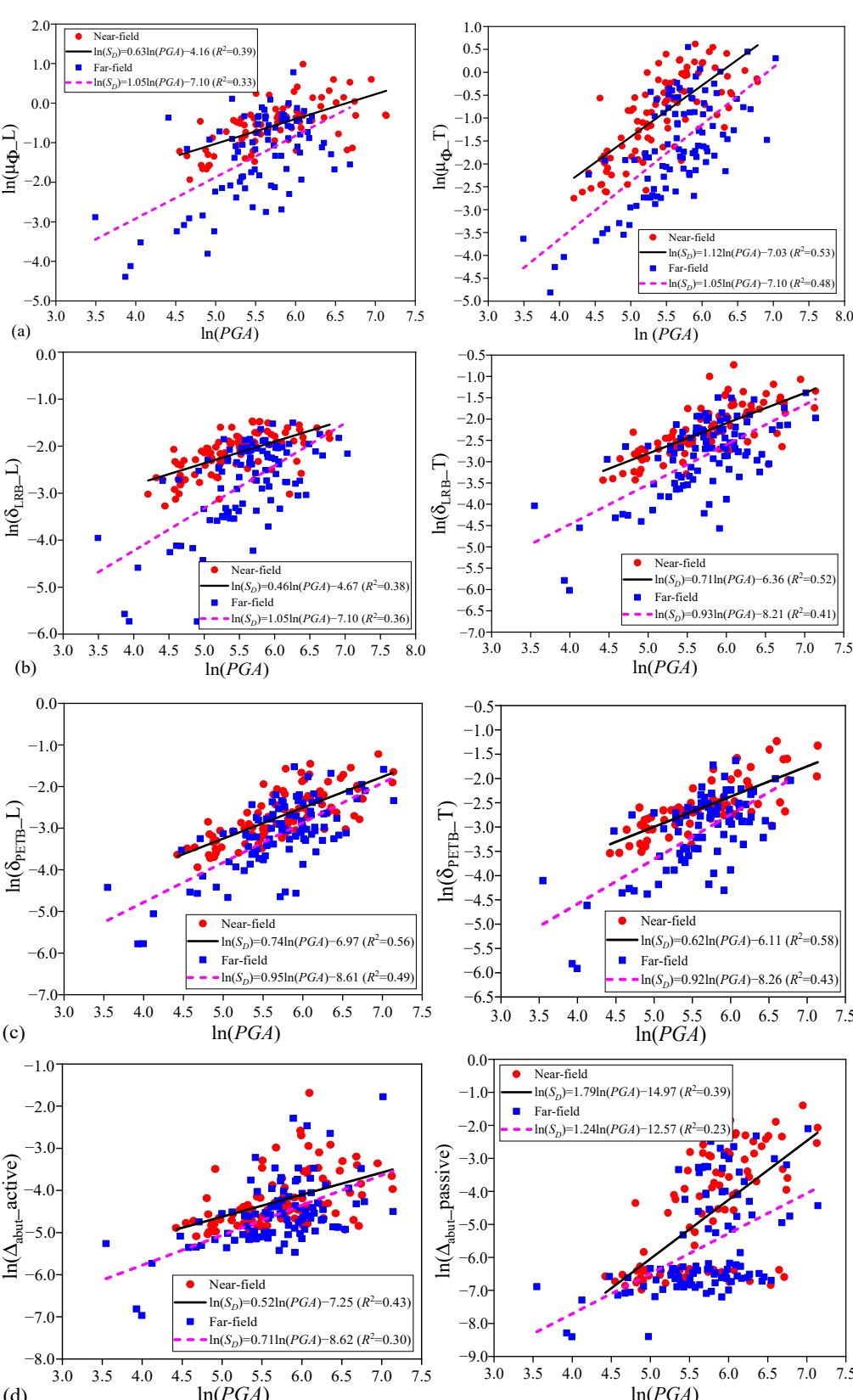

**Figure 11.** Comparison of the developed PSDMs for different bridge components under the near-field and far-field ground motions: (**a**) pier; (**b**) LRB; (**c**) PETB; and (**d**) abutment.

Furthermore, according to the authors' previous study [5], the PSDMs can be used to reflect the relationship of the EDPs and the seismic IM. To investigate the effects of selection

of ground motion IMs on the developed PSDMs, Figures 12 and 13 show the comparative studies of the obtained PSDMs for $u_{\Phi\_L}$ and $\delta_{\text{LRB\_L}}$ under both the near-field and far-field ground motions by using *PGA* and *SA* as the ground motion IMs, respectively. As seen from Figures 12 and 13, it is concluded that the IM selection can be helpful in reducing the influence of uncertainty stemming from the RTR variability of ground motions on the seismic response prediction results (the dispersions of the developed PSDMs). For example, when using *PGA* as the IM, $\beta_{D\,|\,IM}$ values of the developed PSDMs of $u_{\Phi\_L}$ under the near-field and far-field ground motions are 0.45 and 0.73, respectively, whereas those for the developed PSDMs of $u_{\Phi\_L}$ using SA are 0.30 and 0.24, respectively. Similar results can be also found for $\delta_{\text{LRB\_L}}$.

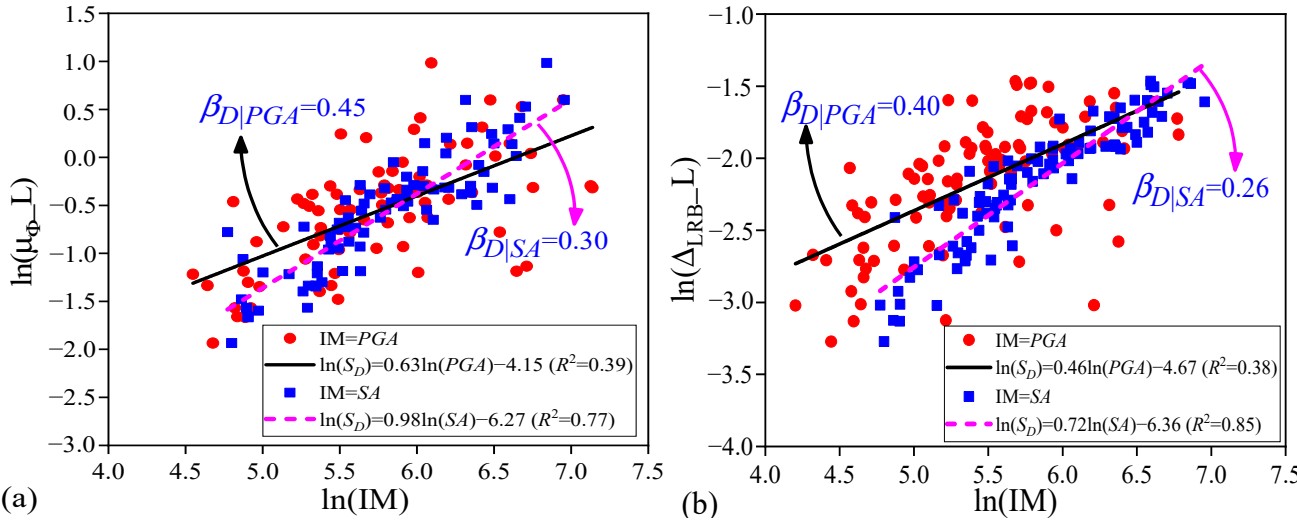

**Figure 12.** Comparison of the developed PSDMs for different bridge EDPs under the near-field ground motions using *PGA* and *SA* as the seismic IM: (**a**) $u_{\Phi\_L}$; (**b**) $\delta_{\text{LRB\_L}}$.

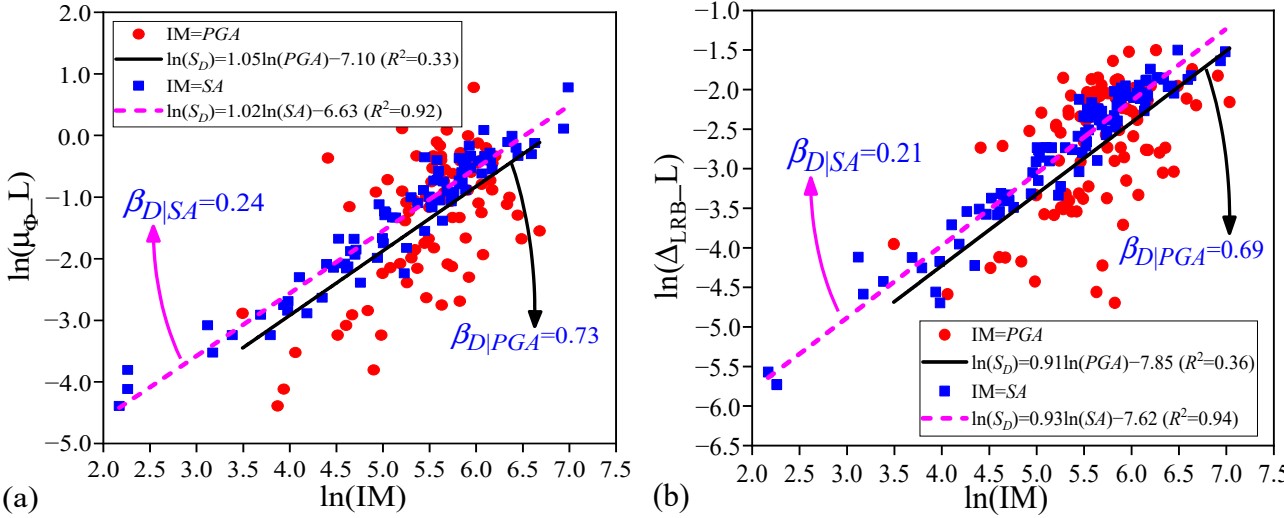

**Figure 13.** Comparison of the developed PSDMs for different bridge EDPs under the far-field ground motions using *PGA* and *SA* as the seismic IM: (**a**) $u_{\Phi\_L}$; (**b**) $\delta_{\text{LRB\_L}}$.

## 8. Conclusions

This study investigates the suitability of 28 commonly used seismic IMs for conditioning the developed PSDMs of a typical RC continuous girder bridge under both the near-fault and far-field seismic records. NTHAs are carried out to generate the PSDMs for different bridge EDPs in predicting the seismic responses of critical bridge members

under both the far-field and pulse-like near-fault ground motions. In addition, influences of the ground motion-related uncertainty stemming from both the bin-to-bin (BTB) and record-to-record (RTR) variabilities of the input seismic records on the PSDA of highway bridges are investigated through the developed PSDMs. Finally, it can be concluded that:

(1)   IM efficiency is the most important criterion in reflecting the RTR variability of ground motions. For both the far-field and pulse-like ground motions, efficiency of the ground motion IMs that is related to structural form, spectrum-related ground motion IMs (i.e., *SA* and *CSA*), and velocity-based IMs (i.e., *PGV* and *VSI*) is good and helpful in reducing the RTR variability of ground motions. An efficient IM will reduce the influence of the RTR variability of ground motions in the predictions of structural demands.

(2)   Both the BTB and RTR variabilities of ground motions have important effects on the PSDA and the developed PSDMs of highway bridges. On the one hand, uncertainty stemming from the BTB variability of ground motions may lead to a significant difference in the developed PSDMs, so it is necessary to carefully select the input seismic records in the PSDA of highway bridges. On the other hand, for a given selected ground motion bin or database, uncertainty derived from the RTR variability of seismic records can also result in discreteness of the PSDMs.

The present study investigated the effectiveness, efficiency, practicality, proficiency, and sufficiency of a number of ground motion IMs and evaluated the ground motion-related uncertainties (e.g., BTB and RTR variabilities) in the PSDA of highway bridges. Based on the acquired analysis results, it is found that the BTB and RTR variabilities of ground motions have significant effects on the developed PSDMs of bridge structures. Therefore, in the future seismic analysis of highway bridges, it is necessary to incorporate the ground motion-related uncertainties to achieve a more fundamental and robust seismic design of highway bridges, as well as other critical lifeline infrastructures, which are also required in the United Nations' 2030 Agenda for Sustainable Development.

**Author Contributions:** Conceptualization, H.L. and J.W.; methodology, H.L.; software, H.L.; validation, H.L., G.Z. and J.W.; formal analysis, H.L.; investigation, H.L. and G.Z.; resources, H.L.; data curation, H.L.; writing—original draft preparation, H.L.; writing—review and editing, H.L.; visualization, G.Z.; supervision, H.L. and J.W.; project administration, J.W.; funding acquisition, J.W. All authors have read and agreed to the published version of the manuscript.

**Funding:** This research received no external funding.

**Institutional Review Board Statement:** Not applicable.

**Informed Consent Statement:** Not applicable.

**Data Availability Statement:** Not applicable.

**Acknowledgments:** The research described in this paper was supported by the National Natural Science Foundation of China (51908016).

**Conflicts of Interest:** The authors declare that they have no known competing financial interests or personal relationships that could have appeared to influence the work reported in this paper.

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
