# Peer review of "Selection of Ground Motion Intensity Measures and Evaluation of the Ground Motion-Related Uncertainties in the Probabilistic Seismic Demand Analysis of Highway Bridges"

_buildings, doi:10.3390/buildings12081184_

Round 1
Reviewer 1 Report
The article focuses on a structural probabilistic seismic demand analysis (PSDA) within a performance-based earthquake engineering approach. More precisely, the authors discuss the selection of appropriate intensity measures and the influences of ground motion-related uncertainties on the PSDA and the developed probabilistic seismic demand models. The manuscript contains a lot of valuable information. The discussed topic is actual and of indisputable practical importance. However, I would suggest the authors address the following remarks.
Please define the term “bin” (employed in the text, for example, in the expression bin-to bin). Does it differ from the term employed in https://doi.org/10.1016/j.engstruct.2019.109899?
Please state clearly the novelties in your research compared to previous studies.
Please check equation 4, more precisely, the numerator under the radical.
Regarding the FE model, do you employ the ready-to-use models in the software, or do you enrich the available models? Probably, you have discussed this in your previous publications. In my opinion, however, you should include this information in the present publication. In this context, you could enhance the manuscript by discussing the nonlinearities in the mentioned nonlinear time-history analyses.
Authors state: “The seismic responses of the pier under the near-field ground motions are significantly lower than that under their far-field counterparts… .” (lines 376-377). This statement either needs further motivation or should be reconsidered.
Reviewer 2 Report
Dear Authors,
The paper “Selection of ground motion IMs and evaluation of ground motions-related uncertainties in the PSDA of highway bridges" is methodologically consistent. However, I would like to encourage considering the more detailed comments below as minor revisions and to improve the manuscript before publication in Buildings Journal.
1. The title should be changed; more attractive, concise, and understandable.
2. I think your paper will be benefit to mention that the research you present contributes to achieve the United Nations' 2030 Agenda for Sustainable Development.
3. Discussion is poor, which should embed the results in previous studies and demonstrate the greatest achievements of this study. You need to improve this section by referring to the latest works to justify and demonstrate your work novelty over the previous works.
Good luck.
